# Combining Novel Biomarkers for Risk Stratification of Two-Year Cardiovascular Mortality in Patients with ST-Elevation Myocardial Infarction

**DOI:** 10.3390/jcm9020550

**Published:** 2020-02-18

**Authors:** Naufal Zagidullin, Lukas J. Motloch, Diana Gareeva, Aysilu Hamitova, Irina Lakman, Ilja Krioni, Denis Popov, Rustem Zulkarneev, Vera Paar, Kristen Kopp, Peter Jirak, Vladimir Ishmetov, Uta C. Hoppe, Eduard Tulbaev, Valentin Pavlov

**Affiliations:** 1Department of Internal Diseases, Bashkir State Medical University, Lenin str., 3, 450008 Ufa, Russia; danika09@mail.ru (D.G.); musina.aisylu@mail.ru (A.H.); zurustem@mail.ru (R.Z.); ishv75@mail.ru (V.I.); tulbaev@gmail.com (E.T.); pavlov@bashgmu.ru (V.P.); 2Department of Surgery, Bashkir State Medical University, Lenin str., 3, 450008 Ufa, Russia; 3Department of Urology, Bashkir State Medical University, Lenin str., 3, 450008 Ufa, Russia; 4Department of Electronics and Biomedical Technology, Ufa State Aviation Technical University, Karl Marx str., 12, 450077 Ufa, Russia; lackmania@mail.ru (I.L.); yogrek2@gmail.com (I.K.); popov.denis@inbox.ru (D.P.); 5Department of Informatics and Robotics, Ufa State Aviation Technical University, Karl Marx str., 12, 450077 Ufa, Russia; 6University Clinic for Internal Medicine II, Paracelsus Medical University, Muellner Hauptstrasse 48, 5020 Salzburg, Austria; l.motloch@salk.at (L.J.M.); v.paar@salk.at (V.P.); k.kopp@salk.at (K.K.); p.jirak@salk.at (P.J.); u.hoppe@salk.at (U.C.H.); 7Institute of Economics, Finance and Business, Bashkir State University, Validy Str. 32, 450076 Ufa, Russia

**Keywords:** myocardial infarction, STEMI, cardiovascular events, cardiovascular death, risk stratification, sST2, NT-proBNP, Pentraxin-3

## Abstract

ST-elevation myocardial infarction (STEMI) is one of the main reasons for morbidity and mortality worldwide. In addition to the classic biomarker NT-proBNP, new biomarkers like ST2 and Pentraxin-3 (Ptx-3) have emerged as potential tools in stratifying risk in cardiac patients. Indeed, multimarker approaches to estimate prognosis of STEMI patients have been proposed and their potential clinical impact requires investigation. In our study, in 147 patients with STEMI, NT-proBNP as well as serum levels of ST2 and Ptx-3 were evaluated. During two-year follow-up (FU; 734.2 ± 61.2 d) results were correlated with risk for cardiovascular mortality (CV-mortality). NT-proBNP (HR = 1.64, 95% CI = 1.21–2.21, *p* = 0.001) but also ST2 (HR = 1.000022, 95% CI = 1.00–1.001, *p* < 0.001) were shown to be reliable predictors of CV-mortality, while the highest predictive power was observed with Ptx-3 (HR = 3.1, 95% CI = 1.63–5.39, *p* < 0.001). When two biomarkers were combined in a multivariate Cox regression model, relevant improvement of risk assessment was only observed with NT-proBNP+Ptx-3 (AIC = 209, BIC = 214, *p* = 0.001, MER = 0.75, MEV = 0.64). However, the highest accuracy was seen using a three-marker approach (NT-proBNP + ST2 + Ptx-3: AIC = 208, BIC = 214, *p* < 0.001, MER = 0.77, MEV = 0.66). In conclusion, after STEMI, ST2 and Ptx-3 in addition to NT-proBNP were associated with the incidence of CV-mortality, with multimarker approaches enhancing the accuracy of prediction of CV-mortality.

## 1. Introduction

Despite the development of new therapeutic strategies, coronary artery disease (CAD) remains one of the main health burdens worldwide. The occurrence of ST-elevation myocardial infarction (STEMI) is especially associated with significant short- and long-term complications. Consequently, STEMI patients are at higher risk of suffering cardiovascular events (CVE), even in a long-term post-myocardial infarction (MI) period, resulting in consequent reduction of long-term survival in this population [1]. Therefore, early identification of high-risk individuals is one of the main clinical goals in daily clinical practice in these patients.

The use of biological markers has been shown to improve the accuracy of diagnosis in cardiovascular patients. Indeed, this approach promotes stratification of cardiovascular risk, both during the hospitalization period as well as in the long-term observation period. Levels of several biomarkers correlate with the severity of CVE, reflect the dynamics of disease and enhance the efficacy of therapy regimes. “Classic” biomarkers like myoglobin fraction of creatine phosphokinase (CK-MB) and Troponins correlate with the long-term outcome of STEMI patients and are integrated into daily clinical practice [1]. Indeed, high levels of N terminals pro brain natriuretic peptide (NT-proBNP) are prognostic for increased risk of sudden death, recurrence of MI or development of chronic heart failure, not only in patients with MI, but also in patients with unstable angina [2]. Nevertheless, with the exception of Troponins and especially high-sensitive Troponins (hs-Troponins), sensitivity and specificity of these biological markers of acute cardiac damage remains poor [3,4,5,6]. Therefore, additional tools are needed to promote the estimation of cardiovascular outcome.

Multimarker analytic approaches have been shown to enhance the sensitivity and specificity of prognostic assessments. Consequently, they might be a more effective tool in predicting cardiovascular mortality (CV mortality) in MI patients. “Novel” serum biomarkers like ST2 and Pentraxin-3 (Ptx-3) have recently emerged as a potentially useful tool for improving the assessment of cardiovascular disease [7,8,9,10,11]. Ptx-3 refers to the family of pentraxins produced locally by stromal and myeloid cells in response to proinflammatory signals. As a multifunctional protein, Ptx-3 plays an important role during vascular inflammatory processes. Consequently, it was shown to have a special role in the pathophysiology of atherosclerosis and myocardial infarction. Furthermore, it also seems to be involved in the pathology of heart failure and cardiac arrest [8]. Indeed, increased Ptx-3 levels are associated with CAD including acute coronary syndrome [7,9,12,13]. Importantly, in patients with acute coronary syndrome, elevated Ptx-3 levels were associated with a higher rate of mortality, even in long-term observational studies [14,15,16]. Soluble ST2 is a member of the of the interleukin 1 receptor family. Its role in cardiac pathophysiological processes including the progression of coronary atherosclerosis but also other cardiac remodeling processes was established in recent years [17]. Indeed, ST2 seems to not only participate in cardiovascular response to injury but also in myocardial remodeling processes observed in heart failure and MI [10,18]. Serum levels are associated with ischemic damage and remain high, even in the post-myocardial infarction period [19]. Consequently, serum concentrations of ST2 correlate with the outcome of MI and also heart failure patients [20,21,22,23,24]. 

Nevertheless, despite these promising results, the ability of both biomarkers to assess the outcome in MI patients still remains the matter of debate. Importantly, the question of whether combining classic biomarkers like NT-proBNP with the new biomarkers ST and Ptx-3 in a multimarker analytic approach in order to improve predictive sensitivity as well as specificity of CV mortality risk in STEMI patients remains unresolved.

Therefore, we investigated serum levels of NT-proBNP as well as ST2 and Ptx-3 in 147 STEMI patients to address this issue and evaluate mid-term cardiovascular outcome. During a two-year follow-up period (FU), initial serum concentrations were correlated with the incidence of CV mortality.

## 2. Methods

In this prospective, non-randomized, single-center study, we enrolled 156 consecutive patients between September 2016 and June 2017, who were hospitalized due to acute STEMI in a cardiac center of Ufa City Hospital N21 in the Russian Federation capable of performing 24/7 percutaneous catheter intervention (PCI) service. Initial diagnosis was established by twelve lead ECG at admission. ST-segment elevation was measured at the J-point at least in two contiguous leads with ST-segment elevation of 2.5 mm in men <40 years, 2 mm in men 40 years, or 1.5 mm in women in leads V2–V3 and/or 1 mm in the other leads in the absence of left ventricular hypertrophy or left bundle branch block. The diagnosis was verified during clinical FU by further ECG recordings (day two and/or day three of hospital stay), transthoracic echocardiographic (day two or day three of hospital stay), laboratory (hs-Troponin I and CK-MB at admission and during FU at day two and/or day three of hospital stay) and coronary angiography according to the 2017 ESC guidelines [1]. Dependent on the time window of STEMI diagnosis, acute coronary angiography (CAG, CAG was performed if time window estimated by primary care physician for possible primary PCI was ≤120 min) or acute thrombolysis (if time window from symptom presentation was ≤12 h and lack of contraindications for thrombolytic therapy was established by primary care physician) were performed. If patients presented with signs of failed fibrinolysis, or if there was evidence of re-occlusion or re-infarction with recurrence of ST-segment elevation indicating unsuccessful thrombolytic therapy, rescue PCI was performed as soon as possible (Table 1). Acute medical treatment, including antiplatelet regime and discharge medication after MI, were established according to the ECS guidelines (Table 1) [1]. Establishment of further relevant diagnoses was performed according to medical history, clinical findings, ECG, laboratory work up and transthoracic echocardiography.

The study was performed in accordance with standards of good clinical practice and the principles of the Declaration of Helsinki. The study was approved by the Ethic committee of the Bashkir State Medical University (N1 from 23 January 2017). Prior to inclusion, all participants signed an informed consent. 

The inclusion criteria were: Age > 18 years and diagnosis of STEMI according to the current guidelines (see above). The exclusion criteria were: > 48 h from start of typical symptoms of acute coronary syndrome (ACS), severe valvular dysfunction defined as severe regurgitation or stenosis of one or more of the cardiac valves, dilative cardiomyopathy, permanent atrial fibrillation and/or atrial flutter, AV block II-III according to medical history and ECG, implanted pacemaker, acute pulmonary embolism, active malignant disease defined as achieved tumor free survival under three years, severe chronic obstructive pulmonary disease (GOLD 2009 stage III-IV), uncontrolled bronchial asthma (according to Global Initiative for Asthma, GINA 2019), acute infectious diseases at the time of STEMI defined as acute pyelonephritis, community acquired pneumonia, acute bronchitis and/or flu/acute respiratory viral infection, and kidney failure defined as glomerular filtration rate (GFR) <30 mL/min1.73 m^2^, as well as pregnancy or lactation.

Patient enrollment and the design of the study are presented in Figure 1. At the day of hospital admission, patients’ venous blood was drawn, subsequently centrifuged and the serum was frozen for further analyses. The concentration of the biomarkers NT-proBNP, ST2 and Ptx-3 was analyzed by enzyme immunoassay as indicated by the manufacturer (for NT-porBNP: Critical diagnostics, USA, for ST2: Biomedica, Slovakia and for Ptx-3: Hycult biotech USA). In addition to the investigated biomarkers, we also evaluated the levels of hs-Troponin I and CK-MB at admission which were routinely measured to verify the diagnosis of STEMI in our center. The serum levels were investigated with the help of the electrochemiluminescence technology for immunoassay analysis as indicted by the manufacturer (Colbas e411, Roche Diagnostics, Switzerland for hs-Troponin I and CK-MB). 

A detailed medical history was obtained at admission for all enrolled patients, including current clinical symptoms, as well as history of previous illnesses, current medications and any further relevant information. The study was carried out between September 2016 and August 2019. Follow-up analysis was conducted over two years ± four months (734.2 ± 61.2 d) from STEMI for the study endpoint with the help of the distant data approach “ProMed” program. The program in the region enables distant online monitoring of hospitalization discharge notes including death certificates. In case of absence of any notes, the patient was contacted by phone at the end of the study period to prevent loss of information due patient relocation to a region where “Promed” was not available.

The study endpoint was defined as cardiovascular mortality (termed CV mortality in this manuscript) as indicated by discharge notes and/or death certificate during the follow-up period. Patients suffering from early death during the first week of acute hospitalization for STEMI were excluded from the analysis. Furthermore, patients suffering from non-cardiovascular deaths (traumas, tumor, cancer, suicides, etc.) and patients lost to FU were also excluded. Consequently, nine patients had to be excluded from the statistical analyses. Three patients suffered from non-cardiovascular deaths during the FU (two deaths were due to trauma incidence and one patient died of cancer disease) while two patients died within one week of acute hospitalization for STEMI. Furthermore, four patients were lost to FU due to relocation and were also excluded from the analyses (Figure 1). Added together, the dropout rate was 5.8% (9/156 patients).

The mathematical model for the statistical analyses is summarized in Figure 2. The statistical analysis was carried out by our blinded statistical analytic team using SPSS software package 21 and R Studio. Data are presented as mean values (M) and standard deviation (SD) for normal distributed variables as well as interquartile range for not normal distributed variables. Mann-Whitney test was used as statistical criteria for determining differences in subgroups as having the greatest statistical power among non-parametric tests with small sample sizes. Qualitative characteristics were analyzed using the standard statistical test Chi-square. To assess cut-off points of biomarkers ROC analysis was used. Kaplan-Mayer survival curves were created after assessment of cut-off points. Log-rank and Gehan’s Wilcoxon tests were applied to estimate CV-mortality and to assess the prediction ability of risk factors. To estimate the quality of multivariate proportional hazard (Cox) regression models and the prognostic ability, measure of explained randomness (MER) and measure of explained variation (MEV) were calculated. The values were estimated without nondependent variables and complete log partial likelihoods function were applied. A *p*-value <0.05 was regarded as statistically significant. 

## 3. Results

Table 1 presents the characteristics of the study population as well as the in-hospital treatment and discharge therapy regime. In summary, men (*n* = 118) prevailed over women (*n* = 29). Patients presented with typical comorbidities observed in the CAD population. If manageable, in-hospital treatment and discharge regime was performed according to current ECS guidelines, as indicated above [1]. 112 patients underwent primary PCI with a success rate of 97.1% (109/112), 35 patients were treated by acute thrombolytic therapy. The success rate of the thrombolytic regime was 48.6% (17/35). Consequently, in this group, 18 patients underwent rescue PCI with a successful rate of 94.4% (17/18). In summary, we observed an overall success of PCI in 126 of 130 treated patients (96.9%). 

Table 2 presents the levels of routinely assessed STEMI relevant cardiac biomarkers.

During a two-year FU (734.2 ± 61.2 d), CV mortality was registered in 33 (22.1%) patients. Scatter plot of the investigated biomarkers with associated survival rates are presented in Figure 3. 

The statistical analysis of the cohort was performed according to the described mathematic model. According to FU and rates of CV mortality, means of ROC analysis cut-off values for the investigated biomarkers were estimated for CV mortality (Table 3, Figure 4). Of note, log-rank and Gehan’s Wilcoxon tests showed significant difference in survival functions between under and upper cut-off value for NT-proBNP (>2141 pg/mL, χ^2^ = 24.0, *p* < 0.001 and χ^2^ = 23.8, *p* < 0.001), ST2 (>27.2 ng/mL, χ^2^ = 14.7, *p* < 0.001 and χ^2^ = 14.3, *p* = 0.022) and Ptx-3 (>169 ng/mL, χ^2^ = 7.0, *p* = 0.001 and χ^2^ = 7, *p* = 0.001). 

Based on our cut-off values, the number and proportion of CV mortality/non-CV mortality (patients surviving) were evaluated (Table 4). The mean concentration of biomarkers in these subgroups are presented in Table 5. 

We created Kaplan-Mayer survival curves for the incidence of CV mortality during the two-year FU comparing under and over cut-off values for the investigated biomarkers NT-ProBNP, ST2 and Ptx-3 (Figure 5). Indeed, for the incidence of CV mortality they showed prominent discrepancies in survival between under and over curve death frequency especially for the biomarker Ptx-3. 

In the next step, the endpoints of the investigated biomarkers were analyzed by univariate Cox regression. NT-pro-BNP and Ptx-3 were analyzed with linear logarithmic and ST2 in quadratic forms. Table 6 presents coefficients of univariate Cox regression for the investigated biomarkers for CV mortality. The Efron approximation of partial likelihood method was used to estimate coefficients of mortality in the Cox model. Indeed, in univariate Cox regression all investigated biomarkers (ST2, NT-proBNP and Ptx-3) were able to predict CV mortality. Of note, in this model, Ptx-3 (univariate Cox model) showed the highest hazard ratio (HR), suggesting that this biomarker may be the most accurate single marker approach for prediction of two-year mortality after STEMI. 

Using Gehan’s Wilcoxon and log-rank tests, patient characteristics (Table 1 and Table 2) were analyzed to asses control variables which are associated with two-year CV mortality (*p* < 0.1; Appendix A). The following variables were shown to be associated with CV mortality during a two-year FU with *p* < 0.1: NT-proBNP, ST2, Ptx-3, age > 65 years, left ventricular ejection fraction (LVEF) < 60% on transthoracic echocardiography, male gender and high level of hs-Troponin I (Appendix A). 

Biomarkers NT-proBNP, ST2 and Ptx-3 were binarized and transformed into dummy-variables, according to cut-off points, obtained above in ROC-analysis. This was done to estimate combined effects of risk factors on CV mortality in two-year FU in a relatively small amount of source data. Also, discrete variables enable more accurate interpretation of hazard ratio in the Cox model.

In the next step, the predictive power of single and multimarker approaches (different combination possibilities of NT-proBNP, ST2 and Ptx-3) were compared for CV mortality on the base of Akaike (AIC) and Schwarz (BIC) information criteria with control variables. The biomarker variables NT-proBNP, ST2 and Ptx-3 were binarized for both models. Furthermore, to confirm our results, MER and MRV values were calculated. One and two-biomarker approaches (ST + NT-proBNP, ST2 + Ptx-3 and NT-proBNP + Ptx-3) and the three-biomarker model were compared to find the most accurate combination according to the AIC and BIC information criteria as well as MER and MEV. 

Table 7 presents the results of coefficients and multivariate risk by Cox model for CV mortality in the two-year FU analyses for three-/two-/one-marker models, according to AIC, BIC, MER and MEV parameters. When comparing single marker models, the application of Ptx-3 (AIC = 211, BIC = 217, *p* < 0.001, MER = 0.69, MEV = 0.56, Table 7) showed the best predictive accuracy of two-year CV mortality, as also indicated by the multivariate regression model. On the other hand, when using ST2 alone, less predictive accuracy was observed (AIC = 220, BIC = 226, *p* = 0.002, MER = 0.49, MEV = 0.39, Table 7). After adding NT-proBNP (AIC = 212, BIC = 217, *p* < 0.001, MER = 0.68, MEV = 0.57, Table 7) or Ptx-3 (AIC = 217, BIC = 222, *p* < 0.001, MER = 0.52, MEV = 0.40), the quality of the model was enhanced, when compared to the application of ST2 alone. However, only minimal improvement was achieved when these two-biomarker models were matched with the single biomarker approach using NT-proBNP or Ptx-3 alone (Table 7). Nevertheless, compared to all investigated single and two biomarker approaches, the combination of NT-proBNP and Ptx-3 (AIC = 209, BIC = 214, *p* = 0.001, MER = 0.75, MEV = 0.64) demonstrated the most powerful quality parameters, indicating this approach to be the most accurate for the prediction of two-year mortality after STEMI, when only two biomarkers are available. Of note, the most accurate combination for prediction of CV mortality after STEMI was observed with the three-biomarker model (AIC = 208, BIC = 214, *p* < 0.001, MER = 0.77, MEV = 0.66, Table 7). Consequently, our results indicate this multimarker approach using the “classic” biomarker NT-proBNP together with the “novel” biomarkers ST2 and Ptx-3 to be a promising tool for the evaluation of the risk for CV mortality during two-year FU after STEMI.

## 4. Discussion

STEMI still represents a leading cause for cardiovascular morbidity and mortality worldwide and is thus also a considerable economic factor [1]. While STEMI patients show high in-hospital mortality rates, they are also at high risk for major adverse cardiovascular events and CV mortality after the acute phase [1,24]. Accordingly, the identification of high-risk patients after STEMI represents one of the main clinical goals. However, despite the evident need, tools for risk-stratification and prognosis after STEMI remain scarce, thereby giving rise to numerous investigations. Of note, numerous studies have proposed a multi-marker approach as best practice. Furthermore, to maximize diagnostic power, a combination of biomarkers from different pathogenetic backgrounds is suggested [11,25,26]. 

In this study, we therefore aimed to evaluate two novel cardiac biomarkers, sST2 and Ptx-3 along with the established cardiac marker NT-pro-BNP for risk stratification in STEMI patients during two-year FU. Of note, all three evaluated biomarkers represent different pathophysiological backgrounds, yet they seem to be of prognostic value for prediction of the outcome in patients suffering from myocardial infarction and associated pathologies like heart failure. 

Indeed, NT-pro-BNP secreted by cardiomyocytes, constitutes a marker mainly utilized in the diagnosis and monitoring of heart failure patients [27]. However, NT-pro-BNP was also shown to be elevated in MI, showing a correlation with the extension of the infarct scar [28,29]. Given its application in routine heart failure FU, the prognostic impact of NT-pro-BNP in STEMI patients was anticipated in previous studies [27,29,30,31]. Ptx-3, a member of the group of pattern-recognition receptors is a marker, involved in the immune-system [7]. Its regulative function in complementary system activation has been considered as a possible mechanism involved in tissue damage after coronary ischemia and reperfusion [8]. Indeed, in larger epidemic studies, this protein was proposed as a prognostic tool showing a significant relationship with cardiovascular and all-cause mortality [32,33]. On the other hand, ST2 represents a marker of inflammation and cardiac stress. There are two known isoforms of ST2, a membrane bound ST2L and a soluble form, sST2 [10]. A ligand to both isoforms is interleukin-33 (IL-33), which is known to mediate cardioprotective effects on a molecular level through binding to the ST2L receptor [34]. In contrast, sST2 acts as a decoy receptor, binding IL-33 and making it unavailable for cardioprotective signaling through the ST2L receptor [34]. Accordingly, an increase in sST2 indicates a decrease in cardioprotective effects. Consequently, this biomarker is elevated in numerous cardiovascular pathologies, such as in heart failure, but also in myocardial infarction [10]. Indeed, several studies have demonstrated its predictive potential in patients suffering from MI [22,35,36]. However, ST2 in isolation cannot be considered as a risk factor. Its low specificity in relation to endpoints during MI was confirmed in the CLARITY-TIMI study [37]. However, as indicated by a subanalysis of this trial, when ST2 is combined with NT-proBNP, the predictive prognostic power of short-term risk stratification in this population is enhanced [37]. Nevertheless, the prognostic power during a longer observation period has not yet been evaluated.

In our trial, all tested biomarkers (ST2, Ptx-3, NT-pro-BNP) showed a promising potential for the prediction of two-year CV mortality after STEMI (Table 6 and Table 7). Nevertheless, when comparing the predictive accuracy using a single marker approach, both in univariate cox regression but also in the multivariate regression model, Ptx-3 levels were associated with the highest accuracy for prediction of two-year CV mortality (Table 6 and Table 7). Of note, these results are in accordance with previous data. Indeed, in MI patients (including STEMI), elevated Ptx-3 levels at hospital admission were associated with higher rate of mortality, even in long-term observational studies [14,15,16]. Furthermore, as already suggested by the CLARITY-TIMI results, in our trial ST2 levels at admission were associated with less predictive accuracy when matched with the other two evaluated biomarkers [37]. We were, however, inspired by further promising data from a subanalysis of CLARITY-TIMI [37] suggesting improvement of short-term risk stratification when combined with NT-proBNP. We decided to investigate its potential value when applied in multimarker models (NT-proBNP+St2 or Ptx-3+ST2). However, when matched with the application of NT-proBNP or Ptx-3 alone, only minimal improvement of predictive accuracy was observed (Table 7). Our results therefore suggest that ST2 may be of lesser value for prediction of midterm (two-year) CV mortality after STEMI when used alone but also when applied in two-biomarker approaches. Interestingly, in our study, the two-biomarker combination of NT-proBNP and Ptx-3 was able to improve the accuracy of the investigated risk assessment (Table 7). Indeed, our results are in accordance with previous trials, which revealed good predictive power for both NT-proBNP but also Ptx-3 when applied in mid- and long-term risk assessment in patients suffering from MI [14,15,16,27,29,30,31]. Nevertheless, to the best of our knowledge, our results indicate for the first time that the combined multimarker approach (NT-proBNP+Ptx-3) might be a promising tool for the prediction of two-year mortality after STEMI, when only two biomarkers are available. Despite promising results revealed by the investigation of this two-biomarker model, in our trial the most accurate combination was observed using a three-marker combination (NT-proBNP+Ptx-3+ST2, Table 7). Therefore, our results might suggest that this strategy is the most promising when utilized for midterm (two-year) risk assessment for CV mortality in a high-risk population. Indeed, these findings support previous speculations, proposing a combination of cardiac biomarkers from different pathogenetic backgrounds for improvement of risk stratification in different cardiovascular pathologies [11,21,25,38].

In our study, the CV mortality rate of 22.1% during two-year FU seems high. However, when compared to previous registry results, it is only slightly higher than cardiovascular endpoint rates in the average European population [39]. Our finding may be mainly attributed to the large rural regions with an inadequate access to medical care and FU system, also represented by the high number of patients undergoing primarily thrombolytic therapy (35/147, 23.8%) in our study. Additionally, high alcohol consumption, an unhealthy diet, a high incidence of metabolic syndrome and social-economic factors must be taken into account in this regard [40,41]. Nevertheless, when exploring high-risk populations, one must also consider potential associated advantages. Indeed, high-risk patients can be effectively identified retrospectively in the described population. Furthermore, our data emphasize the potential need for application of multimarker approaches in populations at increased risk for cardiovascular events.

Compared to previous trials, the cut-off levels for biomarkers proposed in our study are relatively divergent. Indeed, ST2 was shown to be a prognostic marker in the follow-up of heart failure patients, with a cut-off level of >35 ng/mL indicating a worse prognosis [21,23]. However, the calculated cut-off for ST2 for our study endpoint CV mortality was 27 ng/mL. The potential reasons for this finding might be diverse. First, different ELISA kits for ST2 are available, differing substantially in the results for ST2. Second, given the inclusion of patients up to 48 h after onset of symptoms, a delay in blood sampling may be a potential confounder in this regard, with ST2 levels peaking about 6–18 h after onset of symptoms in myocardial infarction [22,35]. Additionally, the young mean age of our study collective may have had an influence on the relatively low cut-off. Nevertheless, while dealing with CV mortality in a STEMI population with higher LVEF, considering the proposed cut-off value of 35 ng/mL in the FU of stable heart failure with reduced ejection fraction patients in numerous studies, our cut-off value seems reasonable [21,23]. On the other hand, regarding NT-pro-BNP, given the time between blood sampling and onset of symptoms as well as the ongoing secretion of NT-pro-BNP following myocardial infarction, these findings also seem adequate. When dealing with Ptx-3, one must consider the lack of a standardized test. Therefore, a nominal comparison to other studies should be considered invalid.

In conclusion, our study proposes a significant correlation of ST2, Ptx-3 and NT-pro-BNP with two-year CV mortality in STEMI patients. All three biomarkers have shown prognostic efficacy in prediction of two-year CV mortality. Nevertheless, when using a single marker approach at admission, the highest accuracy might be associated with Ptx-3 levels, with ST2 levels showing the lowest accuracy. When applying a two-biomarker approach in this setting, the most appropriate two-marker model seems to be the combination of the biomarkers NT-proBNP and Ptx-3 with associated improvement of risk assessment. In our trial, the three-biomarker model (NT-proBNP + ST2 + Ptx-3) was able to predict CV mortality with the highest accuracy indicating this approach to be a promising clinical tool in this risk population. This confirms previous suspicions, suggesting multimarker approaches for risk stratification and monitoring in cardiovascular diseases.

Our study suffers from several limitations. One of the limitations is the relatively small sample size investigated in a single center study. Furthermore, the dropout rate was relatively high (9/156 patients, 5.8%). One must also consider the high rate of thrombolytic therapy (35/147 patients, 23.8%), which is explained by longer patient transfer from distant rural regions to the cardiac center. While hs-Troponin I and CK-MB were applied to confirm diagnosis of STEMI, hs-Tropinin T levels were not routinely used to facilitate the diagnosis of acute MI. As already mentioned, given the inclusion of patients up to 48 h after onset of symptoms, a delay in blood sampling may be a potential confounder. However, our data represent a real-life scenario, which in daily clinical practice is often characterized by various time points of presentation after STEMI. Also, notably, fast and routine applications of measurements of ST2 and Ptx-3 levels are currently still lacking. Therefore, despite promising results, routine application of the proposed multimarker approaches may be limited.

## Figures and Tables

**Figure 1 jcm-09-00550-f001:**
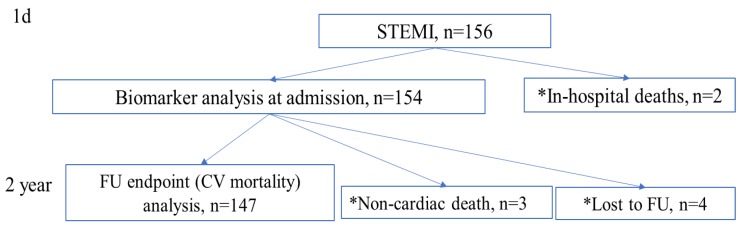
Patient enrollment and the design of the study. *—patients were excluded from the study. FU—follow-up, STEMI—ST-elevation myocardial infarction.

**Figure 2 jcm-09-00550-f002:**
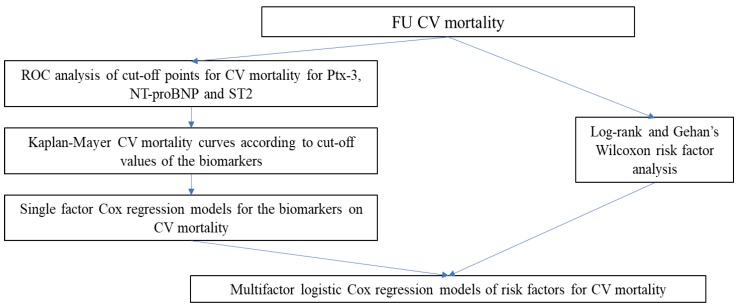
Mathematical model of the statistical analyses. CV mortality—cardiovascular mortality, FU—follow-up, NT-proBNP—N-terminal-pro hormone B-type natriuretic peptide, ROC—receiver operator characteristics, ST2—suppression of tumorigenicity 2.

**Figure 3 jcm-09-00550-f003:**
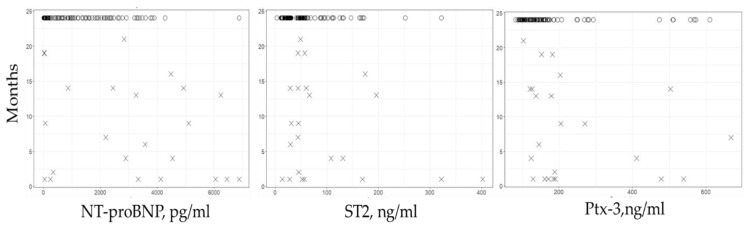
Scatter plots of the investigated biomarkers with associated survival rates (indicated in month) during a two-year FU after STEMI. Cases of CV mortality are indicated by a cross while cases of non-CV mortality are indicated by a circle.

**Figure 4 jcm-09-00550-f004:**
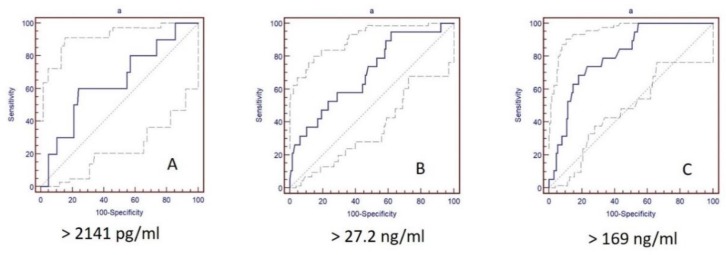
CV mortality cut-off values of the biomarkers NT-proBNP (**A**), ST2 (**B**) and Ptx-3 (**C**) in two-year FU after STEMI by ROC analyses.

**Figure 5 jcm-09-00550-f005:**
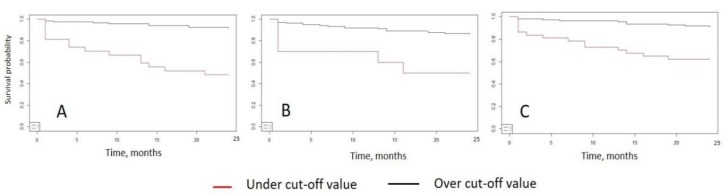
CV Kaplan-Mayer survival curves in two-year FU analyses under and over cut-off values for NT-pro-BNP (**A**), ST2 (**B**) and Ptx-3 (**C**).

**Table 1 jcm-09-00550-t001:** Characteristics of the study cohort.

Parameter	Value
*n*	147
Gender (male)	118 (80.3 %)
Age	60.9 ± 12.1
LVEF (%)	52.8 ± 7.2
Hx stroke (%)	5 (3.4)
Hx MI (%)	34 (23.1)
Smoker (%)	86(58.5)
Arterial hypertension (%)	138 (93.9)
Dyslipidemia (%)	111 (75.5)
DMT2 (%)	37 (25.2)
Revascularization strategy	
Acute thrombolytic therapy (%)	35 (23.8)
Successful thrombolytic therapy (%)	17 (48.6)
Acute thrombolytic therapy followed by rescue PCI (%)	18 (51.4)
Acute PCI only (%)	112 (76.2)
Successful PCI (%)	126 (96.9)
Target vessel in acute/rescue PCA: LCA (%) LAD (%) CX (%) RCA (%) Multivessel approach (%)	1 (0.7) 51 (38.1) 12 (8.9) 48 (35.8) 12 (8.9)
Discharge medication	
ACE inhibitors/Angiotensin receptor blockers *n* (%)	143 (97.3)
Beta-blockers (%)	139 (94.6)
Diuretics (%)	51 (34.7)
Aldosterone antagonists (%)	37 (25.2)
Ivabradine (%)	12 (8.1)
Statins (%)	139 (94.6)
Acetylsalicylic acid (%)	142 (96.0)
Thienopyridines (%)	138 (93.8)
Warfarin	1 (0.7)
NOAK (%)	7 (4.8)

ACE—angiotensin converting enzymes inhibitor, CX—circumflex artery, DMT2—diabetes mellitus type 2, Hx—history of, LAD—left anterior descending artery, LCA—left main coronary artery, LVEF—left ventricle ejection fraction, NOAK—new oral anticoagulants, PCI—percutaneous coronary intervention, RCA—right coronary artery, ST2—suppression of tumorigenicity 2.

**Table 2 jcm-09-00550-t002:** Patients’ investigation data.

Parameter	Median (Q1, Q3)
*n*	147
CK-MB, mmol/L	100.8; (38, 175)
hs-Troponin I, ng/mL	688.4; (41, 2270)
NT-proBNP, pg/mL	518.5; (54, 2130)
ST2, ng/mL	43.8; (24.8, 56.5)
Pentraxin-3, ng/mL	131.5; (110.8, 164.3)

CK-MB—creatine kinase MB fraction, NT-proBNP—N-terminal-pro hormone B-type natriuretic peptide, ST2—suppression of tumorigenicity 2.

**Table 3 jcm-09-00550-t003:** Biomarker cut-off values for CV mortality in a two-year FU after STEMI (*p* < 0.1).

Biomarker	CV Mortality				
	Cut-Off	Sens. %	Spec. %	AUC	*p*-Value
Ptx-3, ng/mL	>169	68.4	82.0	0.804	0.063
NT-pro-BNP, pg/mL	>2141	73.7	80.5	0.801	0.063
ST2, ng/mL	>27.2	94.7	38.3	0.698	0.071

AUC—area under the curve, CV mortality—cardiovascular mortality, NT-proBNP—N-terminal-pro hormone B-type natriuretic peptide, Ptx-3—pentraxin 3, Sens.—sensitivity, Spec—specificity, ST2—suppression of tumorigenicity 2.

**Table 4 jcm-09-00550-t004:** CV mortality/survivals according to cut-off values in a two-year FU after STEMI (*p* < 0.1).

	NT-proBNP, pg/mL		ST2, ng/mL		Ptx-3, ng/mL	
	>2141	≤2141	>27.2	≤27.2	>169.0	≤169.0
*n*	39	108	97	50	36	111
CV mortality, *n* (%)	14(35.9)	5 (4.6)	18(18.6)	1(2.0)	13(36.1)	6(5.4)
Non-CV mortality, *n* (%)	25(64.1)	103 (95.4)	79(81.4)	49 (98.0)	23(63.8)	105(94.6)

CV mortality—cardiovascular mortality, NT-proBNP—N-terminal-pro hormone B-type natriuretic peptide, Ptx-3—pentraxin 3, ST2—suppression of tumorigenicity 2.

**Table 5 jcm-09-00550-t005:** Concentration of biomarkers in CV mortality/non-CV mortality subgroups in a two-year FU after STEMI presented as mean with SD.

	*n*	NT-proBNP, pg/mL	ST2, ng/mL	Ptx-3, ng/mL
CV mortality	33	3019.0 ± 2270.5	93.7 ± 97.1	236.8 ± 158.5
Non-CV mortality	114	1015.8 ± 972.2	51.3 ± 47.3	158.2 ± 103.6

CV mortality—cardiovascular mortality, NT-proBNP—N-terminal-pro hormone B-type natriuretic peptide, Ptx-3—pentraxin 3.

**Table 6 jcm-09-00550-t006:** Univariate Cox regression for biomarkers and the incidence of CV mortality after STEMI.

Biomarker	Coefficient ± SE	Hazard Ratio	AUC	CI	*p*-Value
Log (NT-proBNP)	0.49 ± 0.15	1.64	0.777	1.21–2.21	0.001
ST2	0.000013 ± 0.000006	1.000022	0.800	1.00–1.001	<0.001
Log (Ptx-3)	1.12±0.32	3.1	0.738	1.63–5.39	0.005

AUC—area under the curve, CI—confidential interval, NT-proBNP—N-terminal-pro hormone B-type natriuretic peptide, Ptx-3—pentraxin 3, SE—standard error, ST2—suppression of tumorigenicity 2.

**Table 7 jcm-09-00550-t007:** Multivariate regression risk factors analysis for the prediction of CV mortality in two-year FU after STEMI.

Biomarker and Cut-Off Value	Coefficient ± SD	Hazard Ratio	95% CI	*p*-Value
ST2 (AIC = 220, BIC = 226, *p* = 0.002, MER = 0.49, MEV = 0.39)				
ST2 > 27.2 ng/mL	1.36 ± 0.57	3.88	1.27–11.84	0.017
Age > 65 years	1.32 ± 0.48	3.75	1.45–9.73	0.006
Male gender	0.52 ± 0.42	1.68	0.70–4.05	0.242
hs-Troponin I	0.43 ± 0.20	1.54	1.25–1.88	0.088
LVEF < 60%	−0.37 ± 0.51	0.69	0.25–1.86	0.460
Ptx-3 (AIC = 211, BIC = 217, *p* < 0.001, MER = 0.69, MEV = 0.56)				
Ptx-3 > 169 ng/mL	1.66 ± 0.44	5.26	2.23–12.36	0.0001
Age > 65 years	1.04 ± 0.467	2.83	1.13–7.09	0.026
Male gender	1.02 ± 0.45	2.77	1.15–6.66	0.022
hs-Troponin I	0.63 ± 0.29	1.88	1.41–2.51	0.021
LVEF < 60%	−0.28 ± 0.45	0.75	0.31–1.84	0.534
NT-proBNP (AIC = 213, BIC = 219, *p* < 0.001, MER = 0.66, MEV = 0.54)				
NT-proBNP > 2141 pg/mL	1.74 ± 0.52	5.67	2.05–15.61	0.0008
Age > 65 years	0.53 ± 0.54	1.70	0.59–4.88	0.322
Male gender	0.36 ± 0.45	1.43	0.59–3.44	0.427
hs-Troponin I	0.29 ± 0.22	1.34	1.07–1.66	0.208
LVEF < 60%	−0.31 ± 0.46	0,74	0.30–1.82	0.507
NT-proBNP + Ptx-3 combination (AIC = 209, BIC = 214, *p* = 0.001, MER = 0.75, MEV = 0.64)				
NT-proBNP > 2141 pg/mL	1.67 ± 0.51	5.32	1.95–14.46	0.001
Ptx-3 >169 ng/mL	1.19 ± 0.44	3.28	1.39–7.73	0.007
Age > 65 years	0.51 ± 0.51	1.67	0.60–4.62	0.326
Male gender	0.12 ± 0.21	1.13	0.91–1.39	0.591
hs-Troponin I	0.44 ± 0.22	1.54	1.23–1.92	0.065
LVEF < 60%	0.08 ± 0.12	1.08	0.96–1.22	0.692
NT-proBNP + ST2 combination (AIC = 212, BIC = 217, *p* < 0.001, MER = 0.68, MEV = 0.57)				
NT-proBNP > 2141 pg/mL	1.79 ± 0.49	5.98	2.29–15.60	0.0003
ST2 > 27.2 ng/mL	1.25 ± 0.58	3.48	1.10–10.99	0.03
age > 65 years	0.81 ± 0.51	2.25	0.83–6.10	0.111
Male gender	0.18 ± 0.22	1.20	0.96–1.49	0.281
hs-Troponin I	0.44 ± 0.23	1.54	1.23–1.96	0.058
LVEF < 60%	0.08 ± 0.12	1.08	0.96–1.22	0.696
ST2 + Ptx-3 combination (AIC = 217, BIC = 222, *p* < 0.001, MER = 0.52, MEV = 0.40)				
ST2 > 27.2 ng/mL	1.05 ± 0.59	2.88	0.91–9.08	0.071
Ptx-3 > 169 ng/mL	1.32 ± 0.44	3.74	1.58–8.86	0.003
Age > 65 years	1.26 ± 0.463	3.53	1.43–8.75	0.006
Male gender	0.14 ± 0.22	1.15	0.92–1.43	0.428
hs-Troponin I	0.44 ± 0.21	1.54	1.25–1.88	0.071
LVEF < 60%	0.09 ± 0.11	1.09	0.98–1.23	0.641
NT-proBNP + ST2 + Ptx-3 combination (AIC = 208, BIC = 214, *p* < 0.001, MER = 0.77, MEV = 0.66)				
NT-proBNP > 2141 pg/mL	1.60 ± 0.49	4.95	1.87–13.17	0.001
ST2 > 27.2 ng/mL	0.99 ± 0.59	2.70	0.84–8.69	0.095
Ptx-3 > 169 ng/mL	1.08 ± 0.44	2.94	1.24–6.99	0.055
Age > 65 years	0.73 ± 0.52	2.08	0.76–5.73	0.155
Male gender	0.11 ± 0.21	1.12	0.90–1.38	0.612
hs-Troponin I	0.43 ± 0.22	1.537	1.23–1.93	0.073
LVEF < 60%	0.07 ± 0.12	1.07	0.95–1.21	0.702

AIC—Akaike information criterion, BIC—Schwarz information criterion, LVEF—left ventricle ejection fraction, MER—measure of explained randomness, MEV—measure of explained variation, NT-proBNP—N-terminal-pro hormone B-type natriuretic peptide, Ptx-3—pentraxin 3, ST2—suppression of tumorigenicity 2.

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
