# Peer review of "Combining Novel Biomarkers for Risk Stratification of Two-Year Cardiovascular Mortality in Patients with ST-Elevation Myocardial Infarction"

_jcm, 2020, doi:10.3390/jcm9020550_

Round 1

Reviewer 1 Report

The authors have satisfatory responded to all reviewer's comments and significantly improved the manuscript. I have no more comments and finf it suitable for publication in JCM.

Reviewer 2 Report

Authors did great job in improvising the manuscript.

Introduction is well explained with references.

Improvisation in methods section is impressive, details provided will be really helpful for the reproducibility of the study.

This manuscript is a resubmission of an earlier submission. The following is a list of the peer review reports and author responses from that submission.

Round 1

Reviewer 1 Report

This is a well-written report of a single-center prognostic study based on novel inflammation-related biomarkers.

Specific comments: 

it is arguable that 2-years FU is a long-term study. Results may be presented in more complete way by adding a table showing CVD/nonCVD mean+-SD biomarker data along with their  scatter plots (crude or logtransformed values); mortality rates in low-high concentration (for individual biomarker) subsets should be provided. There is insufficient information on pentraxin-CVD or MACE connection and mechanism

Noteworthy, several novel manuscripts (2018-19) referring to pentraxins or ST2 are not included in discussion or references.

Limitations: 

I disagree that 22% CVD over 2 years is unusual, pleas refer or quote sources as Kardiol Pol. 2015;73(3):142-58. doi: 10.5603/KP.a2014.0213. Therefore Limitations may be refocused: dropout rate is to be mentioned, and single-center design is definitely a limitation along with significant thrombolysis rate; the rates of successful reperfusion are not provided 

Conclusions:

There is no calculation whether the optimal multimarker model was indeed significantly better than single marker-based, therefore conclusions may require rewording.

Table legends must include explanations of all abbreviations used.

Reviewer 2 Report

Introduction needs to be more elaborated with proper references. 

Insufficient information provided in the methods section. Experimental procedure needs to be well explained for the reproducibility of the study. 

Did the authors performed blinded  analysis of the results?

Authors claim about 3 marker model for prediction of CVD for population in risk, what about feasibility of this 3 marker model, if 1/2 markers in the panel are not available or are limited, then which one will be the most preferred one ?

Reviewer 3 Report

This is an interested study on using multimarker approach for risk stratification in STEMI patients, however some issues should be improved and clarified.

Title "Combining" rather than "Combing"; morover I do not think 2 year follow up is a long-term observation in STEMI patients. Modification of the title is needed. Introduction 1st page line 40, authors should clearly state which of the evaluated biomarkers are cardio-specific. Actually, apart from cardiac troponin other biomarkers lack their cardiospecificity. Especially, in the era of hs-troponin assays. Authors should clarify this issue. Page 2 line 54-55 rather "novel" biomarkers. Biomarkers are not instruments as they are usually measured on instruments (analyzers). Page 2 line 63 combining Page 2 line 67-68 what about troponin? Methods page 2, authors should clearly list which biomarkers they tested on admission. Further they mention CK-MB and troponin but they are not mentioned in the Methods section. The measurement procedures for biomarkers should be better descibed. Results Table 2. Biomarker values should be presented in medians (IQR) as they have not normal distribution. The endpoint defined as CVD is not clear. Do you mean MACE? CVD is usually preceeding MI. Please clarify. Table 3 AUC for ST2 is showing less accuracy for prognosis (AUC<0.7 is very poor) Discussion should be more informative and clear. One of the limitations is lack of hs-cTn measurements. Another important limitations for this multimarker approach is lack of fast, rapid routine measurements for some of these biomarkers. This should be improved accordingly.